# Current Research Status of Azaspiracids

**DOI:** 10.3390/md22020079

**Published:** 2024-02-04

**Authors:** Jiaping Yang, Weiqin Sun, Mingjuan Sun, Yunyi Cui, Lianghua Wang

**Affiliations:** Basic Medical College, Naval Medical University, Shanghai 200433, China; smmu_shenghuayjp@163.com (J.Y.); qinjiao1225@163.com (W.S.); sunmj@smmu.edu.cn (M.S.); yunyicui@163.com (Y.C.)

**Keywords:** azaspiracids, detection, biosynthesis

## Abstract

The presence and impact of toxins have been detected in various regions worldwide ever since the discovery of azaspiracids (AZAs) in 1995. These toxins have had detrimental effects on marine resource utilization, marine environmental protection, and fishery production. Over the course of more than two decades of research and development, scientists from all over the world have conducted comprehensive studies on the in vivo metabolism, in vitro synthesis methods, pathogenic mechanisms, and toxicology of these toxins. This paper aims to provide a systematic introduction to the discovery, distribution, pathogenic mechanism, in vivo biosynthesis, and in vitro artificial synthesis of AZA toxins. Additionally, it will summarize various detection methods employed over the past 20 years, along with their advantages and disadvantages. This effort will contribute to the future development of rapid detection technologies and the invention of detection devices for AZAs in marine environmental samples.

## 1. Introduction

Since the 20th century, human beings have progressively explored and exploited marine resources. The high nutritional value of shellfish, along with their abundance of unsaturated fatty acids, has been a significant factor in attracting human consumption. However, shellfish possess self-protective mechanisms that often lead to the production of toxins and other harmful substances for humans. Additionally, as apex predators in the planktonic food chain, shellfish can accumulate toxins produced by algae [1].

Azaspiracids (AZAs) are a type of polyether toxin (Figure 1) that was initially discovered in an episode of food poisoning in Ireland in 1995 [2]. Several individuals in Ireland exhibited symptoms of diarrhea after consuming mussels, leading to the isolation and identification of these toxins [3]. AZAs were originally named Killary Toxin or KT-3 [4], and their name was later changed to Azaspiracids to better reflect the chemical structural formula. Azaspiracids are known to be produced by the Protists *Azadinium* [5,6] and *Amphidoma* [7,8], which belong to the order Lumbar Flagellate. Furthermore, these toxins have been reported and detected in various shellfish species, such as oysters, scallops, and clams [9,10,11].

Since the identification and isolation of AZA-2 and AZA-3 in 1997 [12], numerous analogs of AZAs have been discovered and purified (Table 1). Among these, *Azadinium dexteroporum* is currently known to produce the highest number of AZA variants, with up to six types reported [13]. The total number of named AZA toxins and their analogs currently stands at AZA-68 [14]. (Table 2).

## 2. Toxin Distribution

Currently, there are reports of cases of AZA poisoning on many continents, including North America [15], South America [16], Africa [17], Europe [18], and Asia [19].(Figure 2).

## 3. Toxicology and Pathology

The acute lethal efficacy LD50 values of oral AZA-1, AZA-2, and AZA-3 in mice were determined to be 443 μg/kg, 626 μg/kg, and 875 μg/kg [20], respectively. The final symptoms of oral administration of AZAs in mice include immobility, sternal lateral immobility, tremors, abdominal breathing, hypothermia, and cyanosis. Although diarrhea is the main toxic sign of human ingestion of seafood contaminated with AZAs, oral exposure from AZA-1 to AZA-3 in mice did not cause significant diarrhea. The lethal dose of intraperitoneal injection in mice targeting AZA-1, AZA-2, and AZA-3 is 200 μg/kg, 110 μg/kg, and 140 μg/kg [12]. The signs and symptoms observed after intraperitoneal injection of purified AZAs in mice include progressive paralysis of the limbs, difficulty breathing, and pre-death convulsions [21]. Regarding the determination of 8 AZA-1 analogs and 12 fragments from the synthesis process of AZA-1, it was found that they have very low or almost no toxic effects compared to AZA-1 itself, indicating that the entire AZA-1 molecule and its stereo-orientation are necessary for exerting toxic effects. Animals subjected to AZA treatment exhibit organ swelling, along with the presence of fat droplets and vacuoles in liver cells [22]. The villi of the small intestine become blunt, accompanied by a reduction in the thickness of the brush-like edge. Additionally, there is a mild to moderate increase in apoptotic cells and infiltrating multi-nucleated cells in the mucosal lamina propria. Depletion of white medullary lymph nodes and lymphocyte necrosis are observed in the spleen. Hepatocellular necrosis is evident in liver cells. Prolonged exposure to small amounts of AZA toxins can lead to a decrease in digestive epithelial cells, increased lipid consumption, and an accumulation of lipofuscin [23]. The data on the oral toxicity of toxins in combination with other toxins indicate that neither the combination with OA (okadaic acid) nor YTX (yessotoxin) showed an increase in toxicity, and no overlapping or synergistic effects were found, only gastrointestinal symptoms were observed [24,25].

There are currently no reports on the long-term effects of AZA toxins on humans, but according to the European Food Safety Authority (EFSA), the lowest observed adverse effect level for individuals with a body weight of 60 kg is 1.9 μg AZA-1 equivalents/kg, and based on this, the most acute reference dose is calculated to be 0.2 μg AZA-1 equivalents/kg body weight [26].

## 4. Toxic Mechanism

Currently, there is a lack of definitive experimental research regarding the therapeutic mechanism of AZAs that would indicate its specific target-blocking properties. However, several studies have demonstrated its ability to influence cell electrical activity by affecting potassium ion channels [27], sodium ion channels [28], chloride ion channels [29], and calcium ion channels [30] (Figure 3). However, the specific mode of action, whether through interactions or intermediates, remains to be further explored. Specifically, its effect on the sodium ion channel is limited to the modulation of a fast sodium channel flow rate, without inducing channel inactivation. However, under the action of high-concentration toxins (200 nM), the proportion of channel inhibition can reach 60%, seriously affecting the process of cell depolarization [28]. AZAs can also inhibit sodium current through Na_v_ 1.6 channels in the presence of glutamic acid [31], indicating that AZA poisoning may be due to its synergistic effect on some metabolites. Regarding the mechanism of potassium ion channels, it has been reported that AZAs can inhibit hERG channels (hERG channels play a very important role in myocardial repolarization) [27], and experimental data have shown that it can affect the quantity of hERG in the membrane in rats [32]. Rats treated with different doses of toxins (11 or 55 μg/kg) experienced partial PR interval prolongation and heart rate changes due to potassium ion channel blockade [32]. Inhibition of Ca^2+^ channels primarily affects the storage channels responsible for intracellular calcium ions, thereby giving rise to neurological symptoms [30].

Additionally, experimental data suggest that AZA-1 may act as a tumor initiator [33]. The induction mechanism here may be related to the abundant production of TNF-α and can induce the expression of early response genes jun B, jun D, c-fos, c-jun, fos B, and fra-1 to achieve tumor induction and occurrence. Repeated administration of AZA in mice led to a significant increase in lung tumors, as well as inducing lymph necrosis in tissues such as the small intestine, spleen, and thymus [34]. Prolonged exposure to AZAs in the environment can result in alterations in cellular cytoskeletons and reduced metabolic activity in human cells [28]. Certain experiments have demonstrated an upregulation of mRNA expression of genes associated with cholesterol synthesis and glycolysis following treatment with AZA toxins, suggesting a potential mechanism for AZAs in modulating cellular metabolism [35].

The gastrointestinal symptoms associated with AZA exposure may arise from alterations in the human intestinal glial system which can impact the integrity of the intestinal barrier. These changes include, but are not limited to, induced neuronal alterations, oxidative stress, disruption of the cell cycle, and an increase in specific enteric glial cell (EGC) markers [36]. Furthermore, the synergistic effects of multiple toxins present in the natural marine environment can enhance the virulence of AZAs [37]. Additionally, AZA-1 has been found to have a partial promoting effect on cell apoptosis and induce an increase in genetic toxicity. In terms of the blood system, AZAs can affect the damage to immune system cell lysosomes, consequently impacting phagocytic function [23].

Heart cells subjected to AZA-1 treatment exhibited heightened levels of apoptotic markers, including caspase-3 and -8, cleavage of PARP, and upregulation of Fas ligands [38]. These molecular changes are reflected at the tissue level, resulting in alterations in arterial blood pressure and deposition of cardiac collagen. Long-term experimental studies have demonstrated that AZAs can induce structural changes in the heart that contribute to heart failure [39] and provoke arrhythmias by modulating ion channels [32] (Figure 3).

In summary, although the activity of toxins as inhibitors of PP [40], kinases, and GPCR or as inhibitors of actin polymerization/depolymerization has been experimentally overturned [41], other experiments have shown their cytotoxicity, affecting cytoskeleton arrangement, promoting tumors, and potentially affecting the activity of multiple ion channels. However, at this point in time, there seems to be no scientific consensus on a specific target or mechanism of AZAs that can jointly explain the various effects observed in experiments and the gastrointestinal symptoms observed in exposed individuals, so further exploration of treatment is needed in the future.

## 5. AZA Analogs from Different Sources

The initial characterization of AZA and its analogs was conducted in 1998 using mass spectrometry (MS) and nuclear magnetic resonance (NMR) techniques [3]. Over 60 types of AZA analogs have been identified, with the majority being produced through metabolic processes in mussels.

AZA-38 and 39 are primarily produced by *Amphidoma languida*, a small *dinoflagellate* species belonging to the *Amphidomataceae* family. Recently, these toxins have undergone structural modifications [42]. The toxins produced by different ribosomal subtypes of *Azadinium spinosum* exhibit variations, with subtype A mainly producing AZA-1 and AZA-2, while subtype B primarily produces AZA-11 and AZA-51 [43]. In the case of ribosomal subtype A of *Azadinium pomorum*, there is either no production or only a minimal amount of AZAs, whereas ribosomal subtype C can generate AZA-40 and AZA-2 [44]. Different strains of *Azadinium pomorum* produce the hydroxylation product AZA-42 from AZA-41 and the dehydrogenation product AZA-62 from AZA-11. The former strain is isolated from the South China Sea, while the latter strain is isolated from the northern coast of Chile [45]. Furthermore, AZA-59 is the sole AZA toxin produced by *Azadinium pomorum* strains isolated from the Pacific northwest coast of the United States.

## 6. Synthesis In Vivo

To date, a majority of studies have pointed to the blue mussel, *Mytilus edulis*, as the primary vector for AZAs. However, other organisms such as mollusks, arthropods, and echinoderms have also been reported as potential vectors [46]. In mussels, several AZAs undergo acyl ester formation, with some studies suggesting that these esters exhibit higher toxicity than the free toxins themselves [47]. The average distribution of AZAs in mussels is as follows: hepatopancreas (60.6%), gills (12.0%), and adductor muscle (27.4%) [48].

AZA has been detected in various bivalve mollusks, including oysters (*Crassostrea gigas*), scallops (*Pecten maximus*), clams (*Tapes filipinarium*), and cockles (*Cardium rule*), as well as in numerous phytoplankton species [10]. In the experiment involving feeding blue mussels (*Mytilus edulis*) with toxins, it was found that AZA-17 and AZA-19 were mainly fast metabolites feeding AZA-1 and AZA-2 [49], indicating that carboxylation of methyl groups at the C22 position is a dominant metabolic pathway, while hydroxylation and decarboxylation are secondary degradation pathways. Notably, AZA-65 and AZA-66 have been identified as intermediate products in the conversion of AZA-1 and AZA-2 to AZA-17 and AZA-19. In mussels, the expression of AZA-1-3 can lead to the production of AZA-8, AZA-12, and AZA-5 through C-23 α hydroxylation. Additionally, the double hydroxylation of AZA-67 and AZA-68 can generate AZA-1 and AZA-2 as secondary metabolites [14].

Experimental evidence has demonstrated a correlation between toxin production in algae and temperature, with the highest toxin concentrations observed at 26 °C [50].

AZA-1 treated with rat liver microsomal extract undergoes oxidation at the F ring and can bind with glucuronic acid at C1 to generate glucuronides [51].

## 7. Synthesis In Vitro

In the synthesis of AZAs, the FGHI ring exhibits relatively high stability in the region spanning from C-26 to C-40, whereas the variability is primarily observed in the C-1 to C-22 [52] region. As early as 2008, the structure of the AZA-1 compound was fragmented and partially synthesized, encompassing the synthesis of the E ring, HI ring, CD ring, and FG ring. The conversion of furan into the ABCD ring can be achieved through a single oxygen-initiated one-pot process [53]. Subsequently, the combination of the EFGHI ring and the ABCD ring enables the complete synthesis of AZA-1 [54]. The artificial in vitro synthesis of fragments from C-22 to C-40 has been enhanced and refined [55]. Currently, through the full synthesis of AZA-1 and continuous modification and optimization of reaction conditions, an artificial stereoisomeric composite closely resembling the structure of natural AZA-3 has been achieved [56].

## 8. Detection of Toxins

Many countries, including those in the European Union (EU), have established regulatory limits for AZA content in shellfish intended for human consumption. In the EU, this limit is set at 160 µg of AZA equivalents per kg of shellfish meat (whole body or any edible part) [57]. Currently, the European official method for detecting AZAs is the mouse bioassay [58]. Evaluating the suitability of chromatographic conditions, multi-experimental groups also provided experimental evidence for the standardized authentication of AZA concentrations and a new CRM from the NRC Certified Reference Materials Program (Halifax, NS, Canada) [59,60,61] (Table 3).

### 8.1. LC-MS

LC-MS, or liquid chromatography–mass spectrometry, is a powerful analytical technique that combines the physical separation capabilities of liquid chromatography (LC) with the mass analysis capabilities of mass spectrometry (MS). When combined, LC separates the chemical components of a mixture, and then, those individual components are funneled into the mass spectrometer. The mass spectrometer further separates the chemicals based on their mass-to-charge ratio, and then identifies and quantifies them. In the field of toxin detection, LC-MS has been intensively employed to generate high-quality data, demonstrating its pivotal role in advancing our understanding of various toxins.

In 1999 [12], an LC/MS method was first used that offered sensitive and specific determination of AZA and its two analogs. This method achieved a detection limit of 50 pg for AZAs, demonstrating a sensitivity approximately 8 × 10^4^ times greater than that of the mouse bioassay. Later, a variety of optimized LC-MS (n) methods were developed successively. A highly sensitive LC-MS method achieved a detection limit of 4 pg and allowed for the simultaneous analysis of multiple types of AZAs [9,80]. Additionally, an on-board LC-MS-MS system was developed for near real-time analysis of phycotoxins in plankton [81]. A variety of AZAs have since been found, including its isomers [13,52,82,83]. Since the first discovery of AZA and its two analogs in 1999, this type of toxin has been detected in various marine coasts and marine products worldwide [16,62,84,85,86,87,88,89,90,91,92,93,94,95,96] (Table 4).

Matrix effects can result in either signal enhancement (increase in the apparent concentration of the analyte) or signal suppression (decrease in the apparent concentration of the analyte). It is a significant concern in LC-MS analysis that greatly influences the accuracy and reliability of toxin detection. After an in-depth investigation of matrix effects, it was found that AZA-1 significantly inhibited the signal in the presence of matrix effects. Subsequently, this issue was greatly improved through the use of a rapid LC-MS method that separated major toxins based on the MS ionization mode [97,98]. Furthermore, LC-MS has facilitated rapid simultaneous separation, monitoring, and quantification of toxins [67,79,99].

#### 8.1.1. UHPLC/HPLC-MS

The utilization of ultrahigh-performance liquid chromatography (UHPLC) coupled with advanced mass spectrometry techniques has transformed the detection and quantification of these toxins. Several research groups have combined this method with other techniques such as ESI [71,100,101] and SPE [70,102,103] and optimized it to achieve the simultaneous detection of multiple toxins [76,77], improve detection limits, reduce matrix effects [75], and enhance the effect of chromatographic gradients [104].

#### 8.1.2. SPE/MSPE+LC-MS

The development and refinement of extraction techniques, particularly SPE [64] and MSPE [75], coupled with LC-MS also have significantly advanced our ability to detect and quantify toxins [66,75,102,103,105]. After validation, optimization can significantly reduce matrix effects [106].

#### 8.1.3. LC-ESI-MS

Sensitive LC–electrospray ionization–mass spectrometry (LC-ESI-MS(n)) methods, including ion trap mass spectrometry [84,86,105,107] and Orbitrap technology [51,71,78,101,108,109], have been developed for the determination of major AZAs and their hydroxyl analogs, enabling the separation of multiple AZAs in a short period of time. The combination of LC-MS with ESI technology allows for the determination of MS dissociation pathways and the differences between CID spectra to be obtained [63,65,68,84,110,111,112]. Subsequently, through a series of practical tests [69,113], the utility of this approach was demonstrated, including its ability to reduce matrix effects [114,115].

#### 8.1.4. LC-HRMS

The use of HRMS was described in combination with passive sampling as a progressive approach to marine algal toxin surveys [116]. This method can also be used in conjunction with the Orbitrap exactive HCD mass spectrometer [78], solid–liquid ultrasound-assisted extraction, and solid–phase extraction [103], and has been implemented in the detection process [47,117].

#### 8.1.5. LC-MS+NMR

The combination of NMR and LC-MS is primarily used for the discovery of new AZAs and the detection of the structure and concentration of the toxins [13,118,119,120,121].

#### 8.1.6. LC-MS + Others

Other methods used in conjunction with LC-MS detection technology include live microscopy, quantitative polymerase chain reaction (qPCR) [94], micro-liquid chromatography–tandem mass spectrometry (micro-LC-MS-MS) [63], selected ion monitoring (SIM), multi-reaction monitoring (MRM) [122], and the use of a gel to selectively capture and release AZAs [123].

### 8.2. MBA and CBA

The detection and quantification of toxins, particularly azaspiracids (AZAs), have been significantly advanced by the development and refinement of bioassay techniques, including mouse bioassay (MBA) [124,125,126,127] and cell-based assay (CBA) [127,128]. Due to cost and ethical considerations, these two methods are currently not mainstream detection techniques. They are typically used in conjunction with other technologies to validate the detection results, although such cases are actually quite rare.

### 8.3. Biosensor

At present, the quantification and identification of AZAs are possible only for those compounds that have available certified standards. The development of biosensor assays for detecting AZAs remains a challenge, with no such assay currently available.

Recently, significant progress has been made in the development of AZA-specific antibodies that have shown binding affinity for several AZA analogs, including AZA-1, 2, 3, and 6 [129,130]. These antibodies have been utilized to design competition and sandwich enzyme-linked immunosorbent assays (ELISAs) with a synthetic fragment of the AZA molecule that is conserved for many analogs.

These developments mark a significant step forward in the detection of AZAs. The hope is that in the near future, these new antibodies will be applied in the development of electrochemical or optical immunosensors for the detection of AZAs.

### 8.4. ELISA

In 2015, Samdal [72] demonstrated an ELISA with a working range of 0.45–8.6 ng/mL and a limit of quantitation for total AZAs in whole shellfish of 57 μg/kg. He also produced a new plate coater, OVA-cdiAZA-1, resulting in an ELISA with a working range of 0.30–4.1 ng/mL and a limit of quantification of 37 μg/kg for AZA-1 in shellfish, in 2019 [74]. ELISA can also be employed in conjunction with gel methods [123]. This method is usually combined with bioassay techniques; two research teams have already implemented this technique [125,131].

### 8.5. Other Immunoassays

As part of the advancements in immune technology, microsphere-based immunoassays [132], magnetic bead (MB)-based direct immunoassays [73], and immunoaffinity chromatography (IAC) columns [133], specifically designed for the purification and concentration of AZAs, have been reported. An immunoassay kit was discovered that provided a more sensitive, specific, and swift approach to determining toxins in total shellfish extracts compared to LC-MS [134].

Due to the extensive research and the establishment of regulatory measures concerning AZAs, most studies focused on detection and quantification are primarily concerned with the safety of samples. At the same time, there are also some studies that utilize reference standards to explore new analogs [3,52,82,83]. Since AZA-1 to 3 were the earliest discovered in this group and constitute a significant portion of the entire toxin group (especially AZA-1 and AZA-2 as precursor compounds), with established regulations in terms of food safety, therefore, in the majority of detection efforts, significant attention is placed on these three toxins or the entire AZA toxin family as the target of analysis [92].

We compared the detection results of our investigation group and found that in contrast to other lipophilic toxins, the levels of AZAs were generally low in the samples. In the majority of the tests, AZAs accounted for a minimal proportion and were far below the relevant regulations for food safety. This also implies that the standardization of this toxin is relatively challenging to extract, and it requires a high sensitivity of the detection techniques in practical applications. The AZP toxin group encompasses a wide range of analogs, making it possible for misidentification of toxin types in certain experiments. Therefore, a more precise differentiation of toxin analogs may be a prospective research direction for future studies.

To date, more than 60 analogs of AZAs have been identified, exhibiting complex diversity and similar structures. Therefore, in order to distinguish them more effectively, gaining a clear understanding of their structure and chemical composition through the utilization of MS and NMR spectroscopy, as well as other identification techniques, has become a focal point in the research of these toxins. Gaining a more detailed and in-depth understanding of toxins and developing faster and more efficient detection techniques can contribute to the early development of commercialized detection kits and standardized testing procedures.

Currently, there are relatively few explicitly regulated AZAs. However, in certain detections, the proportion of unregulated AZAs may be higher than that of the established regulated types—potentially even by an order of magnitude. This fact highlights the underestimation of the total AZA load by current regulatory strategies.

## 9. Conclusions

In summary, AZAs are an important part of toxin research. Currently, many countries and institutions have developed monitoring plans and regulatory measures to ensure the safety of seafood products and to safeguard public health. Continuous in-depth research on monitoring and detection methods is crucial for effectively monitoring AZA pollution and protecting consumers from potential harm. Although progress has been made in understanding the toxicology and occurrence of AZAs, there are still some knowledge gaps. Further research is needed to investigate the long-term health effects and potential risks associated with long-term exposure to AZAs. In addition, more research is needed to elucidate the mechanism of action of AZAs and determine the exact relationship between AZAs and their potential carcinogenicity.

Although the probability of large-scale outbreaks of shellfish toxins is low, considering the toxicity, unpredictability, and significant differences in the abundance of AZA toxins in recent years, as well as the threat to consumers and marine professionals, we still need to establish a complete monitoring and protection system. The design of stable and efficient methods and equipment for detecting toxins is also conducive to promoting the development and utilization of marine resources, promoting the safety of marine fisheries, and promoting human health.

## Figures and Tables

**Figure 1 marinedrugs-22-00079-f001:**
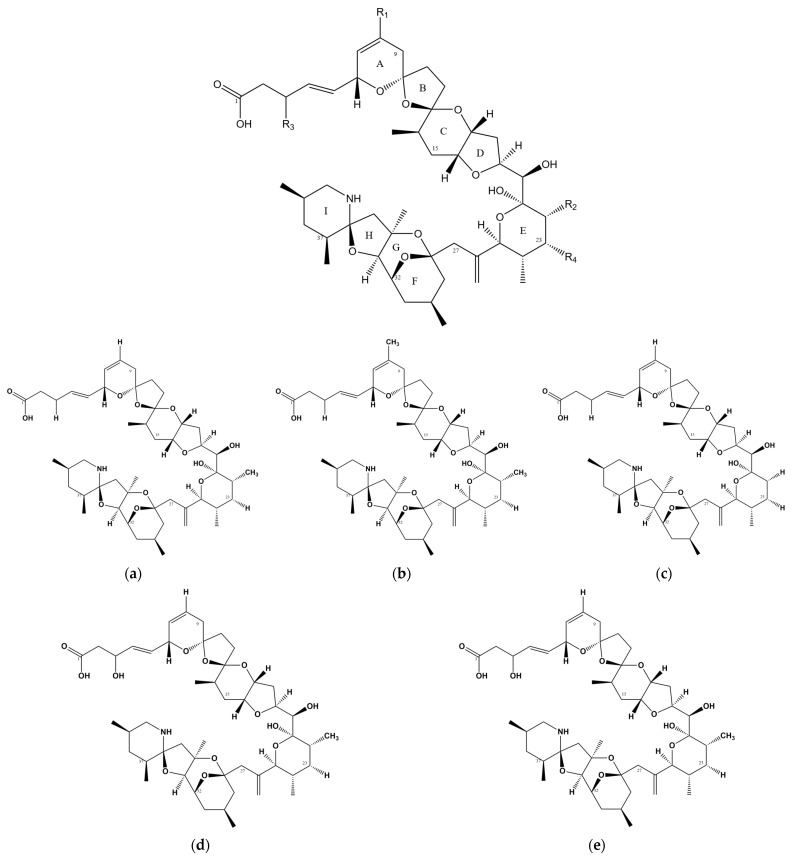
Chemical structure of AZAs. (**a**) Chemical structure of AZA-1, (**b**) chemical structure of AZA-2, (**c**) chemical structure of AZA-3, (**d**) chemical structure of AZA-6, and (**e**) chemical structure of AZA-7.

**Figure 2 marinedrugs-22-00079-f002:**
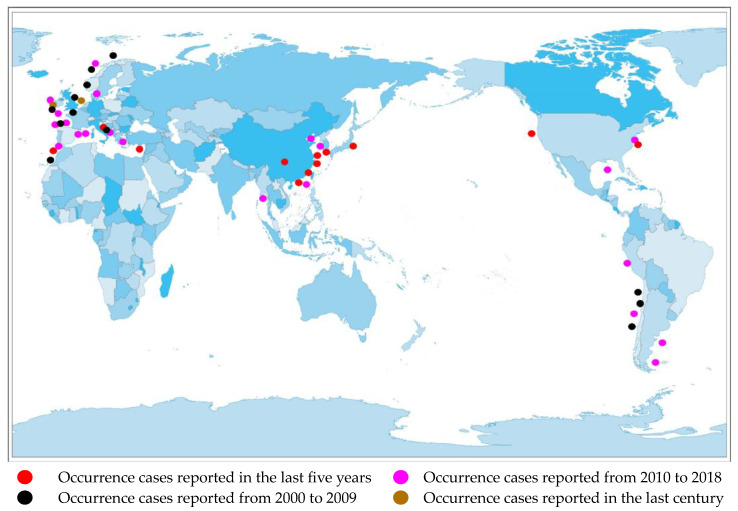
Distribution of AZA separation or occurrence.

**Figure 3 marinedrugs-22-00079-f003:**
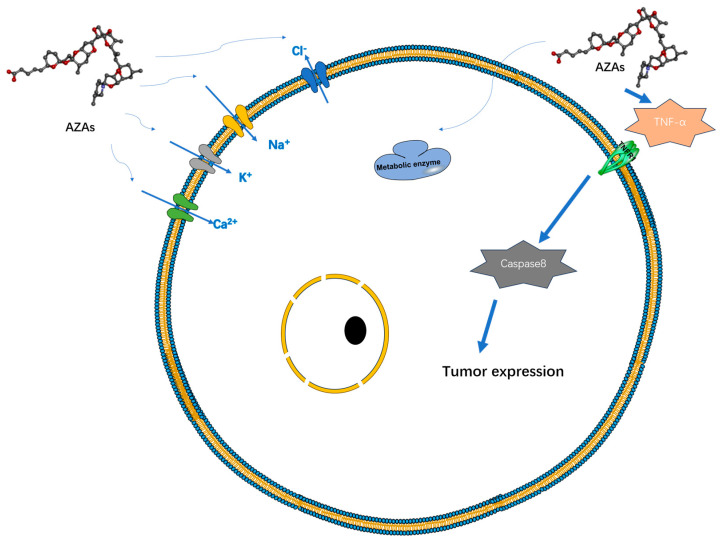
Possible targets and modes of action of AZAs currently reported.

**Table 1 marinedrugs-22-00079-t001:** Chemical structure of some AZA analogs.

Toxins	R_1_	R_2_	R_3_	R_4_
AZA-1	H	CH_3_	H	H
AZA-2	CH_3_	CH_3_	H	H
AZA-3	H	H	H	H
AZA-6	CH_3_	H	H	H
AZA-7	H	CH_3_	OH	H

**Table 2 marinedrugs-22-00079-t002:** Classification of AZA homologous analogs.

AZA-1	AZA-2	AZA-3	AZA-6	AZA-7
AZA-8	AZA-11	AZA-4	AZA-9	AZA-35
AZA-14	AZA-12	AZA-5	AZA-10	AZA-54
AZA-18	AZA-16	AZA-13	AZA-15	AZA-55
AZA-22	AZA-20	AZA-17	AZA-19	AZA-56
AZA-26	AZA-24	AZA-21	AZA-23	AZA-57
AZA-30	AZA-28	AZA-25	AZA-27	AZA-58
AZA-33	AZA-32	AZA-29	AZA-31	
AZA-34	AZA-41	AZA-43		
AZA-36	AZA-42			
AZA-37	AZA-62(AZA-11)			
AZA-38				
AZA-39				
AZA-40				
AZA-50				
AZA-51				
AZA-59				
AZA-63(AZA-37)				
AZA-52(AZA-38)				
AZA-53(AZA-38)				

**Table 3 marinedrugs-22-00079-t003:** LOD and LOQ of different detection methods and sources of samples.

Year	Method	LOD	LOQ	Recovery Rate	Samples
1999	LC-MS [62]	50 pg			Isolated from toxic mussels collected in Ireland
2000	micro-LC-MS [63]	20 ng/g			Isolated from toxic mussels collected in Ireland
2002	LC-MS [9]	4 pg	0.8 ng/mL		Isolated from toxic mussels collected in Ireland
2002	SPE-LC-ESI-MS [64]	5–40 pg	0.05–1.00 μg/mL		Isolated from toxic mussels collected in Ireland
2004	LC-ESI-MS [65]	5 pg	0.05–1.00 μg/mL		Isolated from toxic mussels collected in Ireland
2010	SPE-CID-MS [66]	0.0021 mg/g	0.007 μg/g		From the National Research Council of Canada
2010	SPE-LIT-MS [66]	0.003 mg/g	0.010 μg/g		From the National Research Council of Canada
2010	SPE-HPLC-MS [67]	11.00 pg/g		75.8–82.5%	Extracted from the samples from China
2011	LC-Orbitrap-MS [68]	0.041–0.10 μg/L		96–105%	From the National Research Council of Canada
2014	LC-MS [69]	0.12–13.6μg/kg	0.39–45.4 μg/kg	81.9–119.6%	Extracted from the samples from China
2015	SPE-HPLC-MS [70]	0.013–0.085 μg/kg	1.00 μg/kg	99.2–102%	Extracted from the samples from China
2015	UHPLC-HR-Orbitrap-MS [71]	0.006–0.050 ng/mL	0.018–0.227 ng/mL	96–114%	Isolated from toxic mussels collected in Ireland
2015	ELISA [72]	0.45–8.6 ng/mL	57 µg/kg		Isolated from toxic mussels collected in Ireland
2017	MB-based direct immunoassay [73]	63 μg/kg	120–2875 μg/kg		From the National Research Council of Canada
2019	ELISA [74]	0.30–4.1 ng/mL	37 μg/kg		From the Marine Institute, Ireland
2019	MSPE-UPLC-MS [75]	0.4–1.0 μg/kg	1.0–4.0 μg/kg	82.8–118.6%	From the National Research Council of Canada
2020	SPATT-UPLC-ESI-MS [76]	0.001–0.05 μg/L	0.04 μg/ml		From the National Research Council of Canada
2013	QuEChERS-UHPLC-ESI-MS [77]		0.10 μg/kg	71–108%	Extracted from the samples from China
2014	LC-HRMS [78]		0.9–4.8 pg	80–94%	From the National Research Council of Canada
2020	LC-MS [79]		0.3–0.4 μg/kg	68–129%	From the National Research Council of Canada

**Table 4 marinedrugs-22-00079-t004:** Reported cases of azaspiracid poisoning (AZP) mentioned in the article.

Location of AZP	Year	Area of Production	The Types of Toxins	Results
Ireland	1999 [62]	Arranmore Island	AZA-1-3	Nearly 95% of the total AZAs.
England and Norway	2002 [84]	Craster and Sognefjord	AZA-1-3	61%(AZA-1), 22%(AZA-2), and 17%(AZA-3) (Norway); 72% (AZA-1) and 28% (AZA-3) (UK).
Spain	2007 [85]	Galica	AZA-1	Nearly 15.46% of the total toxins.
Sweden and Norway	2008 [86]	The west coast of Sweden and northwest coast of Norway	AZA-1-3	70.6% (AZA-1) (Sweden); 16.7% (AZA-1) (Norway).
France	2008 [87]	The North Brittany coast	AZAs	80% (AZA-1) and 20%(AZA-2).
Scotland	2008 [88]	The Food Standards Agency, Scotland	AZA-1,2	Nearly 69% of the total samples.
Portugal	2008 [96]	Foz do Arelho beach	AZA-1-3	23.5% (AZA-1), 42.8% (AZA-2), and 33.6% (AZA-3).
Chile	2010 [16]	Coquimbo Bay	AZA-1	Below the quantification limit.
China	2011 [89]	The whole coast of China	AZA-1	Not mentioned.
China	2015 [90]	The main seafood markets along the Chinese coastline	AZA-1	2.75% of the total samples; 27.1% of the total toxins.
Spain	2016 [91]	The North Patagonian coast	AZAs	Not mentioned.
Spain	2017 [92]	The Gulf of Cadiz	AZAs	49.15% (AZA-43) and 50.85% (AZA-2).
The Adriatic Sea	2018 [93]	The coast of Abruzzo and Molise regions	AZA-1-3	23.80% (AZA-1), 42.87% (AZA-2), and 33.33% (AZA-3).
Denmark	2019 [94]	The Limfjord and the Kattegat/Belt area	AZAs	Not mentioned.
Spain	2020 [95]	Galicia	AZA-1-3	Not mentioned.

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
