# Peer review of "Current Research Status of Azaspiracids"

_marinedrugs, 2024, doi:10.3390/md22020079_

Round 1

Reviewer 1 Report

Comments and Suggestions for Authors

In general this is a welcome manuscript about azaspiracids, somewhat neglected toxins that can cause a type of shellfish poisoning. My main concern is that despite the extensive list of literature, the manuscript is somewhat superficial and in certain claims could be better explained. For example the molecular action of AZAs is addressed poorly. All types of ion channels are mentioned but it is not clear if those are  voltage channels or ligand controlled channels. It is not clear from which side AZPs bind to the channels (From Fig. 3 one can conclude that from the inner side), but if so, how the toxins are transported into the cell? Also FIG. 3 is not informative enough and need at least a brief explanation in the figure legend, even more so because the description of these mechanisms in the text  is also very brief and superficial.

In the Introduction the authors are claiming that shellfish protect themselves by toxins production (L.23-24), but shellfish that serve as vectors for algal toxins do not produce their own toxins, except by possible biotransformation of the toxins they acquire by filtration. The algal toxin biotransformation in shellfish is not addressed at all, but this might be an important source that contributes to the variety of AZAs. Due to the structural variants it would be also nice to know which of the variants is the most toxic and what are the ratios of different AZAs found in shellfish samples. The authors should at least address these questions.

L. 102 the ribosomal subtypes of A. spinosum  are mentioned. How are these linked to AZA synthesis and observed variants if at all?

8.5. Better subtitle: Other immunoassays

Comments on the Quality of English Language

The manuscript is generally well written but there are certain paragraphs in which the style should be improved. This would contribute to the overall merit of the text.

Author Response

Jiaping Yang ,  Weiqin Sun ,  Mingjuan Sun

Basic Medical College of Naval Medical University, Shanghai, China

2024.1.13

Dear professor,

Re: Cover Letter for the Manuscript titled "Current Research Status of Azaspiracids"

We are writing to submit our revised manuscript titled "Current Research Status of Azaspiracids" for consideration for publication in Marine Drugs. We appreciate the opportunity to revise our work based on the valuable feedback provided by the reviewers and the editorial team.

1,. the molecular action of AZAs is addressed poorly. All types of ion channels are mentioned but it is not clear if those are  voltage channels or ligand controlled channels. It is not clear from which side AZPs bind to the channels (From Fig. 3 one can conclude that from the inner side), but if so, how the toxins are transported into the cell? Also FIG. 3 is not informative enough and need at least a brief explanation in the figure legend, even more so because the description of these mechanisms in the text  is also very brief and superficial.

Firstly, we would like to address the concerns raised during the review process, regarding the mechanism of action of my toxin, I have added the types of sites and channels of action, as well as the effects it causes.

Secondly, I did overlook the direction of cell membrane action during the drawing process, which resulted in some misleading information. I have made modifications to the image, but further experimental exploration is needed regarding the way toxins enter cells, I am considering adding this part of the experiment in my later experiments to better complete my paper.

2. In the Introduction the authors are claiming that shellfish protect themselves by toxins production (L.23-24), but shellfish that serve as vectors for algal toxins do not produce their own toxins, except by possible biotransformation of the toxins they acquire by filtration. The algal toxin biotransformation in shellfish is not addressed at all, but this might be an important source that contributes to the variety of AZAs. 

Regarding the detoxification mechanism of shellfish toxins, I added in the article the toxic effects of some metabolites of AZA, but their toxicity is much lower than before metabolism, indicating that this is the self-protection mechanism of shellfish.

3. Due to the structural variants it would be also nice to know which of the variants is the most toxic and what are the ratios of different AZAs found in shellfish samples. The authors should at least address these questions.

As for the ratios of different AZAs found in shellfish samples, we have addressed this concern by including the ratios of different AZAs in the detection results, which can be found in Table 4.

4. L. 102 the ribosomal subtypes of A. spinosum  are mentioned. How are these linked to AZA synthesis and observed variants if at all?

Regarding ribosomal typing, it indicates that some subtypes cannot produce toxins, and we should pay attention to distinguishing them when identifying them.

5. 8.5. Better subtitle: Other immunoassays
We have resolved this issue by updating the subtitle to "Other immunoassays," as suggested.

We believe that these revisions have significantly strengthened the manuscript and have addressed the concerns raised during the review process. We would like to express our gratitude to the reviewers and the editorial team for their thoughtful comments and suggestions, which have undoubtedly improved the quality of our work.

We hope that the revised manuscript meets the requirements and standards of Marine Toxin and is suitable for publication. We sincerely appreciate your time and consideration of our work.

Thank you for your attention. We look forward to hearing from you soon.

Jiaping Yang, Weiqin Sun, Mingjuan Sun

Basic Medical College of Naval Medical University, Shanghai, China

Reviewer 2 Report

Comments and Suggestions for Authors

The review covers the discovery, distribution, toxicity, analysis, and chemistry of AZAs. However, there are a few considerations outlined below that lead me to suggest potential improvements for publication in its current form.

1. The subtitle "5. Biosynthesis" may be misleading for this section. A more accurate representation could be "AZA analogues from different sources." This adjustment would better align with the content, as the section doesn't discuss into the intricate details of biosynthetic mechanisms, enzymes, and precursors involved in AZA production in dinoflagellates or shellfish.

2. It is noted that the authors did not reference a significant review paper titled "Azaspiracid Shellfish Poisoning: A Review on the Chemistry, Ecology, and Toxicology with an Emphasis on Human Health Impacts," published in Marine Drugs in 2008. Emphasizing the additional value that the current manuscript contributes to the community in comparison to this prior work would strengthen the discussion.

3. The term "in vitro total synthesis pathway" in the abstract may benefit from clarification. After reviewing the entire paper, it appears that the authors intended:

"In vitro total synthesis" to refer to the chemical synthesis of AZAs in a laboratory setting (in vitro), as opposed to biosynthesis by organisms. This concept is explored further in the section on chemical synthesis methods.

"Pathway" to refer the step-wise process for synthesizing the toxins. However, it's worth noting that "pathway" is more commonly associated with biosynthetic rather than chemical synthesis routes.

4. It is recommended to include structures for other AZA homologues in the manuscript.

Minor issues

1. Maintain consistency in referring to the same Azaspiracid toxin variant. Choose one formatting style, either AZA1 or AZA-1, and use it consistently throughout the manuscript to avoid confusion.

Author Response

Jiaping Yang ,  Weiqin Sun ,  Mingjuan Sun

Basic Medical College of Naval Medical University, Shanghai, China

2024.1.13

Dear professor,

Re: Cover Letter for the Manuscript titled "Current Research Status of Azaspiracids"

We are writing to submit our revised manuscript titled "Current Research Status of Azaspiracids" for consideration for publication in Marine Drugs. We appreciate the opportunity to revise our work based on the valuable feedback provided by the reviewers and the editorial team.

Firstly, we would like to address the concerns raised during the review process.

1. The subtitle "5. Biosynthesis" may be misleading for this section. A more accurate representation could be "AZA analogues from different sources." This adjustment would better align with the content, as the section doesn't discuss into the intricate details of biosynthetic mechanisms, enzymes, and precursors involved in AZA production in dinoflagellates or shellfish.

Regarding the consideration of the subtitle in Part 5, it is indeed as you mentioned. Your suggestion is more reasonable.

2. It is noted that the authors did not reference a significant review paper titled "Azaspiracid Shellfish Poisoning: A Review on the Chemistry, Ecology, and Toxicology with an Emphasis on Human Health Impacts," published in Marine Drugs in 2008. Emphasizing the additional value that the current manuscript contributes to the community in comparison to this prior work would strengthen the discussion.

Regarding the 2008 review, we have conducted research on new things in the past decade, including newly discovered types and mechanisms. Many articles have cited the most original ones, but we have also cited the 2008 review where necessary.

3. The term "in vitro total synthesis pathway" in the abstract may benefit from clarification. After reviewing the entire paper, it appears that the authors intended: "In vitro total synthesis" to refer to the chemical synthesis of AZAs in a laboratory setting (in vitro), as opposed to biosynthesis by organisms. This concept is explored further in the section on chemical synthesis methods. "Pathway" to refer the step-wise process for synthesizing the toxins. However, it's worth noting that "pathway" is more commonly associated with biosynthetic rather than chemical synthesis routes.

Your suggestion regarding the abstract expression has taught me a more rigorous way of expression. In the article, the in vitro synthesis of total synthesis was used, and only the results were cited, The synthesis steps are not well reflected in the article

4. It is recommended to include structures for other AZA homologues in the manuscript.

We have taken this suggestion into consideration and have included structures for other AZA homologues in Figure 1 to provide a comprehensive overview.

5. Maintain consistency in referring to the same Azaspiracid toxin variant. Choose one formatting style, either AZA1 or AZA-1, and use it consistently throughout the manuscript to avoid confusion.

To avoid confusion, we have made the necessary modifications throughout the manuscript. Specifically, we have unified the formatting style for all AZAn, using the format AZA-n consistently.

We believe that these revisions have significantly strengthened the manuscript and have addressed the concerns raised during the review process. We would like to express our gratitude to the reviewers and the editorial team for their thoughtful comments and suggestions, which have undoubtedly improved the quality of our work.

We hope that the revised manuscript meets the requirements and standards of [Journal Name] and is suitable for publication. We sincerely appreciate your time and consideration of our work.

Thank you for your attention. We look forward to hearing from you soon.

Sincerely,

Jiaping Yang ,  Weiqin Sun ,  Mingjuan Sun

Basic Medical College of Naval Medical University, Shanghai, China

Round 2

Reviewer 1 Report

Comments and Suggestions for Authors

I am partially satisfied with your answers to my review, albeit not entirely. First it is unusual that one compound would be able to affect such wide array of different ion channels. Having in mind the diversity of AZA varieties this could be plausible, but cannot be generalized before we know the structure function relation of different AZA and specific type of ion channels. This issue should be addressed with more care, the authors could be more specific about the findings  in the articles cited. In my opinion this information is still too superficial and I believe the graphic (Figure) should have a brief explanation of  events in the legends to the Figure. 

Further if biotransformation of AZAs in vector organism takes place, than one should consider that main AZAs (ie AZA 1, AZA2 and AZA3, respectively) are indeed the most toxic AZAs but then these cannot be or at least may be not involved in human intoxication if they have been already biotransformed in shellfish hepatopancreas to the less toxic metabolites. However, according to the data in Table 4, they were not biotransformed, because AZA1,2 and 3 are the only variants implemented in human intoxication or linked to the toxic shellfish samples. So the question remains what is the origin of so many variants of AZAs and why those are not implemented in human intoxications. This deserves at least a comment and a brief discussion.

Comments on the Quality of English Language

English is OK, minor editing might be required (mostly for typos).

Author Response

Jiaping Yang ,  Weiqin Sun ,  Mingjuan Sun

Basic Medical College of Naval Medical University, Shanghai, China

2024.1.31

Dear professor,

Re: Cover Letter for the Manuscript titled "Current Research Status of Azaspiracids"

We are writing to submit our revised manuscript titled "Current Research Status of Azaspiracids" for consideration for publication in Marine Drugs. We appreciate the opportunity to revise our work based on the valuable feedback provided by the reviewers and the editorial team.

Thank you for your suggestion. Currently, there are only a few compounds that can have an impact on such a large number of ion channels. However, existing literature reports mainly focus on the potential mechanisms of action derived from single-channel research results, where variables are controlled to determine the mechanism of action. Although it has been proven that there is a slight inhibitory effect on many targets, the specific mechanism of action has not been reported or studied extensively.

While I mentioned the existence of AZA analogues, the current experimental subjects mainly consist of AZA1-3. In some cases, these analogues have only shown a reduction in flow rate in certain ion channels, without any observed channel activation or inhibition.

Regarding your second question. All AZA analogues discovered in the past 20 years are based on purification and analysis in organisms other than humans. Currently, there is a lack of separation and determination of AZA's role in causing clinical gastrointestinal symptoms in humans, which presents certain difficulties in identifying the biological transformation of AZA in humans. Additionally, based on observations from mouse experimental results, we have reason to suspect that the gastrointestinal symptoms caused by AZA in humans occur prior to its biological transformation.

We discussed the situation that there are little variations of AZAs in the detection results shown in Table 4 in the article. In recent years, the majority of AZA detection studies have primarily focused on determining whether samples comply with food safety standards. Therefore, in most detection approaches, they utilized standard reference materials for AZA-1, 2, and 3, placing significant emphasis on assessing the compliance of these three toxins' levels with regulatory requirements. Furthermore, a subset of researches did investigate the presence of variants in the samples and compared them to the precursor compounds.

We have made revisions to the article regarding the statement that there is no clear mechanism of action. I hope my revisions can further enhance the expression of the article.

We believe that these revisions have significantly strengthened the manuscript and have addressed the concerns raised during the review process. We would like to express our gratitude to the reviewers and the editorial team for their thoughtful comments and suggestions, which have undoubtedly improved the quality of our work.

We hope that the revised manuscript meets the requirements and standards of Marine Drugs and is suitable for publication. We sincerely appreciate your time and consideration of our work.

Thank you for your attention. We look forward to hearing from you soon.

Sincerely,

Jiaping Yang ,  Weiqin Sun ,  Mingjuan Sun

Basic Medical College of Naval Medical University, Shanghai, China

Reviewer 2 Report

Comments and Suggestions for Authors

The authors have successfully addressed most of the raised concerns, and I have no objections to publication.

Additionally, Figure 2 is missing in the recently submitted manuscript.

Author Response

Jiaping Yang ,  Weiqin Sun ,  Mingjuan Sun

Basic Medical College of Naval Medical University, Shanghai, China

2024.1.31

Dear professor,

Re: Cover Letter for the Manuscript titled "Current Research Status of Azaspiracids"

We are writing to submit our revised manuscript titled "Current Research Status of Azaspiracids" for consideration for publication in Marine Drugs. We appreciate the opportunity to revise our work based on the valuable feedback provided by the reviewers and the editorial team.

First of all, we would like to express our gratitude for your suggestions and recognition of our article. In the final revision, we have made additional modifications and improvements to certain expressions within the manuscript. We believe that these changes will further enrich the content and enhance the overall expression of the article.

Your thorough review and thoughtful comments have undoubtedly enhanced the scientific rigor of our research. We have carefully considered each of your suggestions and incorporated them into the revised version of the manuscript. Your guidance has been instrumental in refining our arguments, clarifying our methodology, and strengthening the validity of our conclusions.

We are sincerely grateful for the time and effort you have dedicated to reviewing our work. Your expertise and attention to detail have significantly contributed to the advancement of our research. We are confident that your feedback will contribute to the overall quality of the publication and its value to the scientific community.

Once again, we extend our heartfelt appreciation for your invaluable support and your vote of confidence in our research. We are honored and grateful to have had the opportunity to benefit from your expertise. Additionally, we would like to thank you for your time dedicated to the review process.

Please do not hesitate to reach out to us if you have any further suggestions or if there are any additional aspects we should address before the final publication. We look forward to the next steps in the publication process.

Thank you once again for your valuable contribution to our manuscript.

We sincerely appreciate your time and consideration of our work.

Jiaping Yang ,  Weiqin Sun ,  Mingjuan Sun

Basic Medical College of Naval Medical University, Shanghai, China

Round 3

Reviewer 1 Report

Comments and Suggestions for Authors

No further questions and remarks.